# Trends in weight gain recorded in English primary care before and during the Coronavirus-19 pandemic: An observational cohort study using the OpenSAFELY platform

Miriam Samuel[1]*, Robin Y. Park[2], Sophie V. Eastwood[3], Fabiola Eto[1], Caroline E. Morton[1,2], Daniel Stow[1], Sebastian Bacon[2], Amir Mehrkar[2], Jessica Morley[2], Iain Dillingham[2], Peter Inglesby[2], William J. Hulme[2], Kamlesh Khunti[5], Rohini Mathur[1], Jonathan Valabhji[6], Brian MacKenna[2], Sarah Finer[1], The OpenSAFELY Collaborative[¶]

1 Wolfson Institute of Population Health, Queen Mary University of London, London, United Kingdom, 2 Bennett Institute for Applied Data Science, Nuffield Department of Primary Care Health Sciences, University of Oxford, Oxford, United Kingdom, 3 MRC Unit for Lifelong Health and Ageing, University College London, London, United Kingdom, 4 Leicester Diabetes Centre, Leicester General Hospital, Leicester, United Kingdom, 5 Diabetes Research Centre, College of Medicine, Biological Sciences and Psychology, University of Leicester, Leicester, United Kingdom, 6 Department of Metabolism, Digestion and Reproduction, Faculty of Medicine, Chelsea and Westminster Hospital Campus, Imperial College London, London, United Kingdom

¶ Membership of the OpenSAFELY collaborative is provided in the Acknowledgements.
* m.samuel@qmul.ac.uk

## Abstract

### Background

Obesity and rapid weight gain are established risk factors for noncommunicable diseases and have emerged as independent risk factors for severe disease following Coronavirus Disease 2019 (COVID-19) infection. Restrictions imposed to reduce COVID-19 transmission resulted in profound societal changes that impacted many health behaviours, including physical activity and nutrition, associated with rate of weight gain. We investigated which clinical and sociodemographic characteristics were associated with rapid weight gain and the greatest acceleration in rate of weight gain during the pandemic among adults registered with an English National Health Service (NHS) general practitioner (GP) during the COVID-19 pandemic.

### Methods and findings

With the approval of NHS England, we used the OpenSAFELY platform inside TPP to conduct an observational cohort study of routinely collected electronic healthcare records. We investigated changes in body mass index (BMI) values recorded in English primary care between March 2015 and March 2022. We extracted data on 17,742,365 adults aged 18 to 90 years old (50.1% female, 76.1% white British) registered with an English primary care practice. We estimated individual rates of weight gain before ($\delta$-prepandemic) and during ($\delta$-pandemic) the pandemic and identified individuals with rapid weight gain (>0.5 kg/m$^2$/

**Data Availability Statement:** Access to the underlying identifiable and potentially re-identifiable pseudonymised electronic health record data is

tightly governed by various legislative and regulatory frameworks, and restricted by best practice. The data in OpenSAFELY-TPP is drawn from General Practice data across England where TPP is the data processor. TPP developers (CB, JC, JP, FH, and SH) initiate an automated process to create pseudonymised records in the core OpenSAFELY database, which are copies of key structured data tables in the identifiable records. These pseudonymised records are linked onto key external data resources that have also been pseudonymised via SHA-512 one-way hashing of NHS numbers using a shared salt. Bennett Institute for Applied Data Science developers and PIs (BG, LS, CEM, SB, AJW, WH, HJC, DE, PI, SD, GH, KB, and CTR) holding contracts with NHS England have access to the OpenSAFELY pseudonymised data tables as needed to develop the OpenSAFELY tools. These tools in turn enable researchers with OpenSAFELY Data Access Agreements to write and execute code for data management and data analysis without direct access to the underlying raw pseudonymised patient data, and to review the outputs of this code. All code for the full data management pipeline—from raw data to completed results for this analysis—and for the OpenSAFELY platform as a whole is available for review at https://github.com/OpenSAFELY. The data management and analysis code for this paper was led by MS and RYP and is available for scientific review and re-use under MIT open licence. https://github.com/opensafely/BMI-and-Metabolic-Markers.

**Funding:** During the period of the analysis MS salary costs salary costs have been supported through a National Institute for Health and Care Research (NIHR) funded academic clinical fellowship in primary care (ACF-2017-19-006) and NIHR grant funding (NIHR AI-MULTIPLY Consortium NIHR203982) https://www.nihr.ac.uk/. There was no direct funding for this study. The funders had no role in the study design, data collection and analysis, decision to publish, or preparation of the manuscript.

**Competing interests:** MS salary costs have been supported through a National Institute for Health and Care Research (NIHR) funded academic clinical fellowship in primary care (ACF-2017-19-006) and NIHR grant funding (NIHR AI-MULTIPLY Consortium NIHR203982) https://www.nihr.ac.uk/. RYP is supported by the EPSRC Centre for Doctoral Training in Health Data Science (EP/S02428X/1). RYP was previously employed as a data scientist for the Bennet Institute which is funded by grants from the Bennett Foundation, Wellcome Trust, NIHR Oxford Biomedical

year) in each period. We also estimated the change in rate of weight gain between the prepandemic and pandemic period (δ-change = δ-pandemic—δ-prepandemic) and defined extreme accelerators as the 10% of individuals with the greatest increase in their rate of weight gain (δ-change $\geq 1.84$ kg/m$^2$/year) between these periods. We estimated associations with these outcomes using multivariable logistic regression adjusted for age, sex, index of multiple deprivation (IMD), and ethnicity. *P*-values were generated in regression models. The median BMI of our study population was 27.8 kg/m$^2$, interquartile range (IQR) [24.3, 32.1] in 2019 (March 2019 to February 2020) and 28.0 kg/m$^2$, IQR [24.4, 32.6] in 2021. Rapid pandemic weight gain was associated with sex, age, and IMD. Male sex (male versus female: adjusted odds ratio (aOR) 0.76, 95% confidence interval (95% CI) [0.76, 0.76], *p* < 0.001), older age (e.g., 50 to 59 years versus 18 to 29 years: aOR 0.60, 95% CI [0.60, 0.61], *p* < 0.001)); and living in less deprived areas (least-deprived-IMD-quintile versus most-deprived: aOR 0.77, 95% CI [0.77, 0.78] *p* < 0.001) reduced the odds of rapid weight gain. Compared to white British individuals, all other ethnicities had lower odds of rapid pandemic weight gain (e.g., Indian versus white British: aOR 0.69, 95% CI [0.68, 0.70], *p* < 0.001). Long-term conditions (LTCs) increased the odds, with mental health conditions having the greatest effect (e.g., depression (aOR 1.18, 95% CI [1.17, 1.18], *p* < 0.001)). Similar characteristics increased odds of extreme acceleration in the rate of weight gain between the prepandemic and pandemic periods. However, changes in healthcare activity during the pandemic may have introduced new bias to the data.

## Conclusions

We found female sex, younger age, deprivation, white British ethnicity, and mental health conditions were associated with rapid pandemic weight gain and extreme acceleration in rate of weight gain between the prepandemic and pandemic periods. Our findings highlight the need to incorporate sociodemographic, physical, and mental health characteristics when formulating research, policies, and interventions targeting BMI in the period of post pandemic service restoration and in future pandemic planning.

## Author summary

### Why was this study done?

- Restrictions imposed during the Coronavirus Disease 2019 (COVID-19) pandemic may have led to lifestyle changes that are associated with weight gain.

- Some studies have suggested that women, younger adults (18 to 29 years), and those living in more deprived areas were at greatest risk of weight gain during the pandemic, but these were limited by small sample size.

- There are currently no large-scale analyses of how the pandemic impacted preexisting patterns of weight gain, and how this varied by sociodemographic and clinical characteristics.

Research Centre, NIHR Applied Research Collaboration Oxford and Thames Valley, Mohn-Westlake Foundation. SVE has been funded by a Diabetes UK Sir George Alberti research training fellowship (grant number: 17/0005588) and University College London Hospitals Biomedical Research Centre, Cardiovascular theme. FE salary cost is supported by MRC (MR/S027297/1) "Multimorbidity, clusters, trajectories and genetic risk in British south Asians, 2020-2023". DS is funded by the NIHR (NIHR203982). AM is a senior clinical researcher at the University of Oxford in the Bennett Institute, which is funded by grants from the Bennett Foundation, Wellcome Trust, NIHR Oxford Biomedical Research Centre, NIHR Applied Research Collaboration Oxford and Thames Valley, Mohn-Westlake Foundation. AM has consulted for https://inductionhealthcare.com/. AM has represented the RCGP in the health informatics group and the Professional Advisory Group that advises on access to GP Data for Pandemic Planning and Research (GDPPR); the latter role is paid. AM is a former employee and Chief Medical Officer of NHS Digital (having left NHS Digital in January 2020). AM has consulted for health care vendors, the last time in 2022; the companies consulted in the last 3 years have no relationship to OpenSAFELY. RM is supported by Barts Charity (MGU0504). JV was the National Clinical Director for Diabetes and Obesity at National Health Service (NHS) England from April 2013 to September 2023 and is funded by the Imperial National Institute for Health Research (NIHR) Biomedical Research Centre and North-West London NIHR Applied Research Collaboration. BMK is also employed by NHS England. KK is supported by the National Institute for Health Research (NIHR) Applied Research Collaboration East Midlands (ARC EM) and the NIHR Leicester Biomedical Research Centre (BRC). KK has acted as a consultant, speaker or received grants for investigator-initiated studies for Astra Zeneca, Bayer, Novartis, Novo Nordisk, Sanofi-Aventis, Lilly and Merck Sharp & Dohme, Boehringer Ingelheim, Oramed Pharmaceuticals, Roche and Applied Therapeutics. SF has received grants from the NIHR (NIHR 31672, NIHR 202635) and MRC (MR/W014416/1, MR/V004905/1, MR/S027297/1). SF, RM, CM are part of the Genes & Health programme, which is part-funded (including salary contributions) by a Life Sciences Consortium comprising Astra Zeneca PLC, Bristol-Myers Squibb Company, GlaxoSmithKline Research and Development Limited, Maze Therapeutics Inc, Merck Sharp & Dohme LLC, Novo Nordisk A/S, Pfizer Inc, Takeda Development Centre Americas Inc. This research used data assets made available as part of the Data

## What did the researchers do and find?

- We used data from the routinely collected health records of 17 million adults living in England to investigate patterns of weight change before and during the pandemic and describe which groups were most likely to have experienced unhealthy patterns of weight gain.

- We found that, among adults with measures of weight recorded in their healthcare records, women, younger adults (18 to 29 years), those living in the most deprived areas, and those of white British ethnicity were most likely to have gained weight rapidly before and during the pandemic.

- The same groups were also most likely to have experienced extreme acceleration in their rate of weight gain during the pandemic.

- Almost all the long-term conditions (LTCs) increased the risk of unhealthy patterns of weight gain, with mental health conditions, such as depression, having the greatest estimated effect.

## What do these findings mean?

- The COVID-19 pandemic appears to have had the greatest impact on women, young adults, and those living in the most deprived areas in terms of unhealthy patterns of weight gain.

- We present, to our knowledge, new evidence that people with mental health conditions were more likely to have unhealthy patterns of weight gain during the pandemic, highlighting the need to consider sociodemographic, physical, and mental health characteristics in research, policies, and interventions around weight.

- We recommend that future pandemic planning should consider how to mitigate the unequal indirect impact of a pandemic on patterns of weight gain to prevent preexisting health inequalities being further exacerbated.

- Changes in healthcare activity during the pandemic affected patterns of weight monitoring in primary care, which may have introduced new bias to the analyses.

## Introduction

Obesity and rapid weight gain are established risk factors for noncommunicable diseases [1] and have emerged as independent risk factors for severe disease following Coronavirus Disease 2019 (COVID-19) infection [2–5]. In March 2020, restrictions imposed to reduce COVID-19 transmission resulted in profound societal changes that impacted many health behaviours, including physical activity and nutrition [3,6–8]. However, the suspension of population health surveys, such as the Health Survey for England, during the pandemic [9] resulted in a lack of quantitative data on weight changes.

A systematic review of observational studies reported a modest increase in adult weight during the pandemic, but analyses were limited by small sample size and non-representative

and Connectivity National Core Study, led by Health Data Research UK in partnership with the Office for National Statistics and funded by UK Research and Innovation (grant ref MC_PC_20058). In addition, the OpenSAFELY Platform is supported by grants from the Wellcome Trust (222097/Z/20/Z); MRC (MR/V015757/1, MC_PC-20059, MR/W016729/1); NIHR (NIHR135559, COV-LT2-0073), and Health Data Research UK (HDRUK2021.000, 2021.0157).

**Abbreviations:** aOR, adjusted odds ratio; BMI, body mass index; CI, confidence interval; COPD, chronic obstructive pulmonary disease; COVID-19, Coronavirus Disease 2019; CVD, cardiovascular disease; EHR, electronic health record; GP, general practitioner; IMD, index of multiple deprivation; IQR, interquartile range; LTC, long-term condition; NHS, National Health Service; SD, standard deviation; SMI, serious mental illness; TIA, transient ischaemic attack.

samples [10]. Previous studies using routinely collected healthcare records have been limited to populations accessing healthcare via specific health providers in the United States of America [11,12], or individuals with long-term conditions (LTCs) [13]. In these settings, women [11], young adults [11], those living in low income neighbourhoods [12], and those with LTCs [12,13], including depression [12], were at greatest risk of weight gain during the pandemic [7]. However, these findings have not been replicated in a population-representative cohort. Additionally, groups such as young adults were at increased risk of weight gain even prior to the pandemic [14]. Therefore, a comparison of individual rates of weight gain, before and after the onset of the pandemic is required to understand the specific impact of the pandemic and identify individuals who had the most extreme acceleration in their rate of weight gain during the pandemic. Body mass index (BMI) data, a measure of weight adjusted for height (BMI = weight in kilograms (kg)/height in metres squared ($m^2$)), recorded in English primary care electronic health records (EHRs) have been shown to provide population level BMI estimates that are comparable to large nationally representative population surveys and have been used to estimate rates of weight gain (BMI trajectories) before the pandemic [14,15].

Using routinely collected English primary care records we conducted an observational cohort study to: (i) describe population-level changes in BMI recorded in healthcare records during the pandemic; and (ii) estimate individual-level rates of weight gain before and during the pandemic to identify how (iia) risk of rapid weight gain ($>0.5$ kg/$m^2$/year) in the pre-pandemic and pandemic periods and (iib) risk of extreme acceleration in the rate of weight gain ($\geq 1.84$ kg/$m^2$/year) between the prepandemic and pandemic period varied by sociodemographic and clinical characteristics.

## Methods

### Data source

Primary care health records, managed by the health record software provider TPP (https://tpp-uk.com/), were linked, stored, and analysed securely within the OpenSAFELY platform inside TPP, which contains pseudonymised data on approximately 40% of the English population, including coded diagnoses, medications, and physiological parameters. No free text data are included. Further details are available at https://opensafely.org/. Detailed pseudonymised patient data is potentially re-identifiable and therefore not shared. The study was approved by the London School of Hygiene & Tropical Medicine (LSHTM) Ethics Board (reference 26536). An information governance statement is included (S1 Appendix).

### Analysis plan

We conducted an observational study following the analysis plan set out in the prospective protocol submitted to the LSHTM Ethics Board (S1 Protocol). The study aimed to evaluate, among adults living in the United Kingdom, how trends in measures of weight have changed since the onset of the COVID-19 pandemic using data routinely collected in primary care EHRs. The analysis was subsequently limited to adults living in England due to data availability through the analysis platform. Due to the size and complexity of the dataset the protocol was also adapted in order to facilitate the analysis within the computational capacity. Specifically, based on literature review and consensus from a clinical advisory panel, we refined the covariates included in the analysis and adjusted the age range of the population studied (from $>18$ to $\leq 110$ years in the protocol to $>18$ to $\leq 90$ years in the completed analysis). This study is reported as per the Reporting of studies Conducted using Observational Routinely collected health Data (RECORD) Statement (S1 Checklist).

## Data management

Data management was done with Python 3.8 and SQL, and analysis was done using R 4.0. Prevalence counts are rounded to the nearest 5 to reduce risk of disclosure. All code for data management and analysis, as well as codelists is shared openly for review and re-use under MIT open license, available at https://github.com/opensafely/BMI-and-Metabolic-Markers.

## Study population

We extracted data on all male and female adults aged $\geq 18$ to $\leq 90$ years who had been registered with a primary care practice in England using EHR software provided by TPP, a healthcare technology provider (S1 Appendix), for at least 1 year prior to 1 March 2022.

## Study outcomes: Body mass index (BMI) trends

We used changes in body mass index (BMI = weight in kilograms (kg)/height in metres squared ($m^2$)), as a proxy for weight change. BMI is recorded in primary care records during routine health checks, disease monitoring pathways for conditions such as diabetes, and when weight or BMI recording is indicated during the primary care consultation based on clinical judgement. We extracted all BMI data using recorded weight (in kilograms) and height (in metres), or directly from recorded BMI values (in $kg/m^2$) between 1 March 2015 and 1 March 2022, from the point an individual reached 18 years of age. Individuals who joined the Open-SAFELY-TPP database after 1 March 2015 had BMI data extracted between their date of first registration with the general practitioner (GP) practice and 1 March 2022. Recordings taken within healthcare settings and self-reported values are included in EHRs and cannot be differentiated using clinical codes. We extracted monthly values per individual and took the most recent in instances where multiple values per calendar month were present. Extreme values (BMI $<15$ $kg/m^2$ and BMI $>65$ $kg/m^2$) were omitted from all quantitative analyses related to BMI values to censor erroneous results and exclude extremely overweight and underweight individuals whose patterns of weight change may not reflect those seen in the general population. Cut off values were decided by consensus of a clinical advisory panel (2 diabetologists and 2 primary care clinicians).

We first investigated population level trends in patterns of BMI recording in primary care and median recorded BMI. We examined the proportion and characteristics of the population having at least 1 BMI value recorded in the year beginning March 2019 (2019), March 2020 (2020), and March 2021(2021) to characterise how the pandemic had influenced BMI data availability. There is no information in the EHR to indicate whether recorded weight, height, and BMI values are recorded in a clinical setting or self-reported. We were therefore unable to characterise how the proportion of self-reported BMIs changed during this time. We then calculated the population-level median and interquartile range (IQR) of the recorded BMI values for each of the years above, using the median per year, where multiple values were available for a single individual. Median (IQR) values are presented rather than means with standard deviation as recommended for data that are not normally distributed.

We then estimated individual level BMI trajectories as rates of BMI change per year ($\delta$) in $kg/m^2$/year after the onset of the pandemic ($\delta$-pandemic). We also present estimates for the rate of BMI change in the prepandemic period ($\delta$-prepandemic) to give context to the pandemic analysis. BMIs were classified into time periods: period-1: March 2015–February 2018; period-2: March 2018–February 2020; and period-3 March 2020–February 2022. For patients with more than 1 BMI measure from each period, a random BMI measure was selected from each period using the slice_sample function from the R dplyr package [16]. Where data were available, BMI data from period 1 and period 2 were used to calculate $\delta$-

prepandemic, while BMI data from period 2 and period 3 were used to calculate δ-pandemic. Rate of BMI change/year was calculated between these time points assuming a linear trend [14] (S2 Appendix).

Through the recommendations of a clinical advisory panel, individuals with known cancer and those underweight (BMI <18.5 kg/m$^2$) in the prepandemic period were excluded from these analyses as rapid weight gain in these groups may have reflected improved disease control or other positive health outcomes. The most extreme 0.05% of values of δ-prepandemic and δ-pandemic (a positive or negative change of >6 kg/m$^2$/year) were censored to reduce the impact of erroneous results. This cut off was defined by a data driven approach and confirmed with input from a clinical advisory panel. Individuals gaining >0.5 kg/m$^2$/year were classified as experiencing rapid weight gain [1]. We estimated associations between sociodemographic and clinical characteristics and rapid weight gain before and during the pandemic using descriptive statistics and regression models (see Statistical models).

To identify individuals who had experienced the greatest acceleration in their rate of weight gain between the prepandemic and pandemic period, we first estimated individual-level changes in rate of weight gain (δ-change) as the difference in the prepandemic and pandemic rates of weight gain (δ-change = δ-pandemic—δ-prepandemic, S3 Appendix). A positive δ-change indicated the rate of weight gain increased or, if patients were losing weight prepandemic, the rate of weight loss during the pandemic became slower than the prepandemic period (e.g., if an individual was losing 1 kg/m$^2$/year prepandemic which slowed to losing 0.5 kg/m$^2$/year during the pandemic). We then defined the 10% of the study population experiencing the greatest acceleration in their rate of weight gain (δ-change ≥1.84 kg/m$^2$/year) as "extreme accelerators." We estimated associations between sociodemographic and clinical characteristics and extreme acceleration using descriptive statistics and regression models (see Statistical models).

## Covariates

Covariates were chosen by clinical consensus, guided by data availability, and known potential predictors of BMI change. Covariates included age (18 to 29 years, 30 to 39 years, 40 to 49 years, 50 to 59 years, 60 to 69 years, 70 to 79 years, 80 ≤ 90 years), sex (female or male); ethnicity recorded in primary care records and categorised according to the 2001 UK Census definitions (white British, white Irish, other white, black African, black Caribbean, other black, Indian, Pakistani, Bangladeshi, Chinese, other Asian, mixed white/black African, mixed white/black Caribbean, mixed white/Asian, other) [17]; deprivation using the most recent patient postcode-derived index of multiple deprivation (IMD) (by quintiles from those living in the most deprived 20% (IMD1) of households to the least deprived 20% (IMD5)); and the presence or absence of the following LTCs ever-recorded: hypertension; type 1 diabetes (T1D), type 2 diabetes (T2D), asthma, chronic obstructive pulmonary disease (COPD), anxiety and depression, serious mental illness (SMI: psychosis or bipolar disorder), learning difficulties, dementia, cardiovascular disease (CVD: conditions affecting the heart), and stroke and transient ischaemic attack (stroke and TIA). Individuals were assigned to age groups based on their age at the following time points: for δ-prepandemic analyses age in March 2015; for δ-pandemic analyses age in March 2018; for δ-change analyses age in February 2022 (S2 Appendix). IMD is an area-based marker of deprivation based on an individual's place of residence [18,17]. Due to the size of the dataset, extraction of LTC data was limited to conditions for which there were established primary care pathways associated with enhanced payments (Quality Outcomes Framework or Pay for Performance). These conditions were chosen for 2 reasons: firstly, they

represent conditions which have been identified as having a significant population health impact, and secondly, as data on the management of these conditions is submitted for enhanced payments the quality of clinical coded data is high [19]. The complete code lists used to define each LTC (hypertension, T1D, T2D, asthma, COPD, SMI, learning difficulties, dementia, CVD, stroke and TIA) in the health record are available for open access and review (https://github.com/opensafely/BMI-and-Metabolic-Markers). Data on all covariates were extracted yearly.

## Statistical models

We used complete case analysis (inclusion of individuals with all baseline covariate data) in all statistical models. This is consistent with previous studies using English primary care data [18,20]. As this is a descriptive study, multiple imputation techniques were not employed as we were not trying to gain an unbiased causal estimate of a single exposure-outcome association, and therefore did not have an analytic model around which to build an imputation procedure. Additionally, the missing at random assumption (required for imputation) was unlikely to hold, e.g., individual IMD linked to residence is less likely to be recorded for individuals in unstable accommodation. All individuals had complete data for age and sex as these were part of the study inclusion criteria, and clinical covariates (such as T2D) which were identified based on the presence/absence of specified codes in the data. Therefore, ethnicity and IMD were the only variables with missing data.

For each of the estimated individual-level outcomes (rapid weight gain prepandemic, rapid weight gain pandemic, and extreme acceleration in rate of weight gain), we compared the baseline characteristics of the population with and without the outcome. We also compared baseline characteristics and outcomes between the complete case sample and the entire population, including those with missing ethnicity and deprivation data, to identify differences in individuals with and without missing data.

We used logistic regression to explore associations between the sociodemographic and clinical covariates and the estimated outcomes. Models were adjusted separately for age, sex, IMD, and ethnicity. We present the results of multivariable models adjusted for age, sex, IMD, and ethnicity. Due to the very large sample size of the study population, 95% confidence intervals (CIs) for the prevalence estimates of each of the individual-level outcomes were often the same as the point estimate. We therefore present the point estimate for prevalence estimates, accompanied by the adjusted odds ratio (aOR, adjusted for age and sex) with 95% CIs for the aOR when a comparison is being made between groups.

## Subgroup analyses

We investigated whether estimated associations between the covariates and extreme acceleration in rate of weight gain persisted in populations stratified by age group (18 to 39 years, 40 to 59 years, and 60 to 79 years), sex, IMD (IMD quintile 1 and IMD quintile 5), and ethnicity (black and South Asian (Bangladeshi, Indian, Pakistani)). The ethnicities were grouped into broader groups (white, black, South Asian, Chinese and other, and mixed ethnicity) for subgroup analyses to reduce risk of disclosure from rare events. Age groups were also collapsed into larger groups (e.g., 18 to 29 and 30 to 39 years collapsed into 18 to 39 years) to reduce the risk of disclosure from rare events.

## Patient and public involvement

OpenSAFELY has a publicly available website through which we invite patients or members of the public to contact us about this study or the broader OpenSAFELY project.

## Results

Data were extracted for 17,742,365 adults meeting study inclusion criteria of whom 50.1% were female and 76.1% were of white British ethnicity (Table 1). Fig 1 demonstrates the population contributing data to each stage of the analysis.

### Population level trends in median BMI

The median recorded BMI increased from 27.8 kg/m$^2$, IQR [24.3,32.1 kg/m$^2$] in 2019 to 28.0 kg/m$^2$, IQR [24.4, 32.6] in 2021 (S1 Table). An increase in median BMI was seen in all subgroups, except the oldest age group (aged 80 to 90), people with dementia, and people with T2D. Although for some groups, such as individuals living in the least deprived IMD quintile, this increase was quite modest (IMD 5: 27.0 kg/m$^2$, IQR [23.9, 30.9] in 2019 to 27.1 kg/m$^2$, IQR [23.9, 31.2] in 2021) (Fig 2 and S1 Table). The proportion and characteristics of the population having a BMI recorded changed during the pandemic. BMI was recorded in 31.30%, 95% CI (31.28, 31.33) of the total study population in 2019, dropping to 18.69%, 95% CI (18.67, 18.70) in 2020 before recovering to 24.99%, 95% CI (24.97, 25.01) in 2021, with higher proportions seen among individuals with LTCs in all the study periods (e.g., T2D: 82.54%, 95% CI (82.46, 82.61) in 2019, 72.18%, 95% CI (72.10, 72.26) in 2021) (S2 Table).

### Individual level BMI trajectory analyses

We investigated individual BMI trajectories to identify the groups at greatest risk of rapid weight gain (>0.5 kg/m$^2$/year) among adults with BMI measures recorded in their EHRs. The average rate of weight gain was 0.06 kg/m$^2$/year (standard deviation (SD): 1.20) during the pandemic and 0.08 kg/m$^2$/year (SD: 1.05) prepandemic (S3 Table). We found 29.20% of individuals gained weight rapidly during the pandemic, compared to 26.92% prepandemic (S4 Table). Odds of rapid weight gain during the pandemic (δ-pandemic) were associated with age, sex, deprivation, ethnicity, and a history of LTCs.

The prevalence and adjusted odds of rapid pandemic weight gain reduced with age, for example, prevalence was 30.08% in 50- to 59-year-olds compared to 44.18% in 18- to 29-year-olds (aOR 0.60, 95% CI [0.60, 0.61], $p < 0.001$). Male sex reduced the adjusted odds (23.90% male versus 32.87% female: aOR 0.76, 95% CI [0.76, 0.76], $p < 0.001$), as did living in the least deprived quintile (26.07% least deprived IMD quintile versus 32.61% most deprived IMD quintile; aOR 0.77, 95% CI [0.77, 0.78], $p < 0.001$). White British people had the highest adjusted odds of rapid weight gain compared to all other ethnicities; in comparison, black Caribbean people had the highest prevalence of rapid weight gain but lower adjusted odds (32.41% black Caribbean versus 29.42% white British: aOR 0.91, 95% CI [0.89, 0.94], $p < 0.001$). When comparing individuals with and without specific LTCs, T2D was the only LTC associated with a reduction in the adjusted odds (20.73%, aOR 0.71, 95% CI [0.71, 0.72], $p < 0.001$). Individuals with all the other studied LTCs had an increased prevalence of rapid weight gain, with the greatest prevalence and adjusted odds among those with learning difficulties (36.34%: aOR 1.10, 95% CI [1.08, 1.12], $p < 0.001$), SMI (35.16%: aOR 1.24, 95% CI [1.22, 1.27], $p < 0.001$), and depression (33.36%: aOR 1.18, 95% CI [1.17, 1.18], $p < 0.001$). Characteristics associated with increased adjusted odds of rapid weight gain before the pandemic were similar (Fig 3 and S4 Table).

To define the population experiencing the greatest increase in their rate of weight gain between the prepandemic and pandemic periods (extreme accelerators), we first estimated the change in rate of weight gain in $n$ = 2,768,695 individuals who had prepandemic and pandemic BMI trajectory data (δ-change = δ-pandemic—δ-prepandemic). We found a wide distribution in δ-change (S3 Appendix), half the population had a slight reduction in their rate of weight

**Table 1. Baseline characteristics of the study population at each stage of the analysis.**

| | Total study population | Population level weight trends in the year beginning: | | | Individual level rates of weight change (δ) | | |
|---|---|---|---|---|---|---|---|
| | | March 2019 | March 2020 | March 2021 | δ-prepandemic | δ-pandemic | δ-change |
| | N (%) | N (%) | N (%) | N (%) | N (%) | N (%) | N (%) |
| Total | 17,742,365 | 5,094,590 | 3,173,970 | 4,422,295 | 3,966,500 | 3,214,155 | 2,768,695 |
| Age group (years) | | | | | | | |
| 18–29 | 3,062,705 (17.3) | 580,335 (11.4) | 409,920 (12.9) | 454,245 (10.3) | 274,540 (6.9) | 267,605 (8.3) | 161,655 (5.8) |
| 30–39 | 3,138,645 (17.7) | 579,610 (11.4) | 385,130 (12.1) | 469,950 (10.6) | 466,285 (11.8) | 341,870 (10.6) | 253,415 (9.2) |
| 40–49 | 2,953,105 (16.6) | 735,120 (14.4) | 417,080 (13.1) | 591,250 (13.4) | 545,760 (13.8) | 415,805 (12.9) | 310,045 (11.2) |
| 50–59 | 3,176,680 (17.9) | 958,860 (18.8) | 558,625 (17.6) | 823,905 (18.6) | 757,830 (19.1) | 603,670 (18.8) | 479,655 (17.3) |
| 60–69 | 2,495,010 (14.1) | 980,770 (19.3) | 578,445 (18.2) | 839,940 (19.0) | 771,890 (19.5) | 642,120 (20.0) | 573,030 (20.7) |
| 70–79 | 1,997,575 (11.3) | 885,645 (17.4) | 561,970 (17.7) | 836,005 (18.9) | 757,405 (19.1) | 635,175 (19.8) | 649,920 (23.5) |
| 80–90 | 918,640 (5.2) | 374,250 (7.3) | 262,795 (8.3) | 406,995 (9.2) | 392,785 (9.9) | 307,910 (9.6) | 340,975 (12.3) |
| Sex | | | | | | | |
| Female | 8,883,720 (50.1) | 2,943,330 (57.8) | 1,871,025 (58.9) | 2,591,225 (58.6) | 2,381,345 (60.0) | 1,898,510 (59.1) | 1,612,850 (58.2) |
| Male | 8,858,645 (49.9) | 2,151,260 (42.2) | 1,302,945 (41.1) | 1,831,065 (41.4) | 1,585,155 (40.0) | 1,315,645 (40.9) | 1,155,845 (41.8) |
| Patient IMD quintile | | | | | | | |
| 1 (most deprived) | 3,356,720 (19.4) | 1,011,785 (20.2) | 636,765 (20.5) | 868,410 (20.1) | 836,480 (21.1) | 686,100 (21.3) | 592,805 (21.4) |
| 5 (least deprived) | 3,240,120 (18.7) | 905,350 (18.1) | 552,035 (17.8) | 777,720 (18.0) | 692,065 (17.4) | 549,495 (17.1) | 472,800 (17.1) |
| Missing | 431,235 (2.4) | 89,270 (1.8) | 65,980 (2.1) | 108,220 (2.4) | - | - | - |
| Ethnicity | | | | | | | |
| White British | 12,299,145 (76.1) | 4,017,955 (81.9) | 2,480,500 (81.1) | 3,446,610 (81.1) | 3,275,985 (82.6) | 2,631,235 (81.9) | 2,306,865 (83.3) |
| White Irish | 93,035 (0.6) | 27,760 (0.6) | 17,460 (0.6) | 24,460 (0.6) | 21,500 (0.5) | 17,590 (0.5) | 15,475 (0.6) |
| Other white | 1,540,215 (9.5) | 293,165 (6.0) | 184,800 (6.0) | 254,570 (6.0) | 211,145 (5.3) | 172,260 (5.4) | 131,960 (4.8) |
| Indian | 501,025 (3.1) | 139,620 (2.8) | 91,605 (3.0) | 129,905 (3.1) | 115,585 (2.9) | 99,750 (3.1) | 82,245 (3.0) |
| Pakistani | 358,585 (2.2) | 111,395 (2.3) | 70,880 (2.3) | 99,095 (2.3) | 99,045 (2.5) | 80,400 (2.5) | 68,000 (2.5) |
| Bangladeshi | 83,300 (0.5) | 25,350 (0.5) | 17,330 (0.6) | 23,755 (0.6) | 21,230 (0.5) | 18,965 (0.6) | 15,025 (0.5) |
| Chinese | 115,825 (0.7) | 15,860 (0.3) | 9,570 (0.3) | 12,910 (0.3) | 10,185 (0.3) | 8,245 (0.3) | 6,395 (0.2) |
| Other Asian | 264,080 (1.6) | 65,660 (1.3) | 44,760 (1.5) | 62,390 (1.5) | 51,865 (1.3) | 46,590 (1.5) | 36,420 (1.3) |
| Black Caribbean | 95,140 (0.6) | 32,595 (0.7) | 21,190 (0.7) | 29,520 (0.7) | 27,170 (0.7) | 23,050 (0.7) | 23,970 (0.9) |
| Black African | 219,880 (1.4) | 49,330 (1.0) | 33,185 (1.1) | 46,435 (1.1) | 37,745 (0.9) | 32,875 (1.0) | 20,330 (0.7) |
| Other black | 89,040 (0.6) | 22,845 (0.5) | 14,490 (0.5) | 21,265 (0.5) | 17,295 (0.4) | 15,020 (0.5) | 11,300 (0.4) |
| White and black Caribbean | 49,930 (0.3) | 13,065 (0.3) | 8,905 (0.3) | 11,655 (0.3) | 10,610 (0.3) | 8,840 (0.3) | 6,960 (0.2) |
| White and black African | 39,065 (0.2) | 8,375 (0.2) | 5,660 (0.2) | 7,720 (0.2) | 6,400 (0.2) | 5,460 (0.2) | 3,935 (0.1) |
| White and Asian | 42,740 (0.3) | 9,240 (0.2) | 6,250 (0.2) | 8,530 (0.2) | 7,055 (0.2) | 5,945 (0.2) | 4,445 (0.2) |
| Other mixed | 82,940 (0.5) | 17,055 (0.3) | 11,705 (0.4) | 15,515 (0.4) | 12,525 (0.3) | 10,795 (0.3) | 7,950 (0.3) |
| Other | 286,925 (1.8) | 57,620 (1.2) | 39,790 (1.3) | 53,955 (1.3) | 41,155 (1.0) | 37,130 (1.2) | 27,420 (1.0) |
| Missing | 1,581,510 (8.9) | 187,700 (3.7) | 115,900 (3.7) | 174,020 (3.9) | 148,055(3.5) | 121,040 (2.9) | 86,840 (3.8) |
| LTC | | | | | | | |
| Hypertension | 3,558,405 (20.1) | 1,837,260 (36.1) | 1,939,115 (38.9) | 2,673,785 (60.5) | 1,637,165 (41.3) | 1,436,070 (44.7) | 1,399,300 (50.5) |
| Type 2 diabetes | 1,231,455 (6.9) | 867,735 (17.0) | 669,295 (21.1) | 887,870 (20.1) | 820,390 (20.7) | 793,770 (24.7) | 805,315(29.1) |
| Type 1 diabetes | 93,505 (0.5) | 59,420 (1.2) | 41,440 (1.3) | 55,640 (1.3) | 55,810 (1.4) | 51,250 (1.6) | 47,910(1.7) |
| Learning disability | 105,855 (0.6) | 61,110 (1.2) | 51,200 (1.6) | 65,265 (1.5) | 54,580 (1.4) | 55,750 (1.7) | 49,865 (1.8) |
| Depression | 3,592,080 (20.2) | 1,233,900 (24.2) | 806,460 (25.4) | 1,140,545 (25.8) | 1,105,540 (27.9) | 887,240 (27.6) | 777,355 (28.1) |
| Dementia | 134,080 (0.8) | 48,460 (1.0) | 40,295 (1.3) | 60,785 (1.4) | 58,670 (1.5) | 47,225 (1.5) | 48,415 (1.8) |
| SMI | 195,450 (1.1) | 116,055 (2.3) | 88,595 (2.8) | 118,010 (2.7) | 100,025 (2.5) | 101,395 (3.2) | 88,375 (3.2) |
| COPD | 531,475 (3.0) | 308,030 (6.0) | 181,610 (5.7) | 261,870 (5.9) | 299,095 (7.5) | 227,935 (7.1) | 245,510 (8.9) |
| Asthma | 2,930,645 (16.5) | 1,040,870 (20.4) | 634,885 (20.0) | 871,240 (19.7) | 939,925 (23.7) | 709,730 (22.1) | 638,485 (23.1) |
| CVD | 1,087,320 (6.1) | 588,970 (11.6) | 407,485 (12.8) | 578,685 (13.1) | 560,030 (14.1) | 487,915 (15.2) | 498,305 (18.0) |

(*Continued*)

**Table 1.** (Continued)

| | | | | | | | |
|---|---|---|---|---|---|---|---|
| Stroke and TIA | 465,645 (2.6) | 226,455 (4.4) | 157,285 (5.0) | 229,915 (5.2) | 219,360 (5.5) | 187,580 (5.8) | 190,500 (6.9) |
| Cancer | 905,475 (5.1) | 348,585 (6.8) | 232,940 (7.3) | 348,225 (7.9) | - | - | - |

N (%): Total number and percentage of individuals in each population subgroup. δ-prepandemic: rate of weight gain before the COVID-19 pandemic. δ-pandemic rate of weight gain during the COVID-19 pandemic. δ-change: change in rate of weight gain between the prepandemic and pandemic period.

COPD, chronic obstructive pulmonary disease; COVID-19, Coronavirus Disease 2019; CVD, cardiovascular disease; IMD, index of multiple deprivation; LTC, long-term condition; SMI, serious mental illness (includes psychosis and bipolar disorder); TIA, transient ischaemic attack.

gain (median δ-change = −0.02 kg/m$^2$/year), while 40% of the population experienced an acceleration of more than 0.27 kg/m$^2$/year. The 10% of the study population with the greatest acceleration in their rate of weight gain after the onset of the pandemic (δ-change ≥1.84 kg/m$^2$/year, S3 Appendix) were classified as extreme accelerators. Consistent with our findings on risk of rapid weight gain, women, younger adults (18 to 29 years), and individuals living in the most deprived quintiles had the greatest odds of extreme acceleration in rate of weight gain. Individuals with T1D had a lower prevalence and adjusted odds (9.38%, aOR 0.87, 95% CI [0.85, 0.90], $p < 0.001$) of extreme acceleration than those without T1D. All other LTCs increased the adjusted odds, with mental health conditions, including depression, SMI, and dementia, having the largest estimated effect (Fig 4 and S5 Table).

## Sensitivity and subgroup analyses

In the sensitivity analysis, we compared our findings from the complete case sample to analyses conducted in the entire population, including those with missing ethnicity and deprivation data, and found our results to be consistent in both groups. In the subgroup analyses, we found that most of the estimated associations between the risk of extreme acceleration in rate of weight gain and age, sex, IMD and LTCs persisted in the analyses stratified by age, sex, ethnicity, and deprivation quintile. But there were some exceptions, e.g., among black individuals, deprivation was not associated with extreme acceleration in rate of weight gain (IMD5 versus IMD1: aOR 1.04, 95% CI [0.92, 1.18], $p = 0.496$), and among those living in the least deprived quintiles, black individuals had a higher adjusted odds than white individuals (black versus white: aOR 1.24, 95% CI [1.10, 1.39], $p < 0.001$). There was consistent evidence that most LTCs increased the estimated risk of extreme acceleration in rate of weight gain, with mental health conditions continuing to have the greatest effect. Associations with T2D were less consistent, a history of T2D continued to have an estimated protective effect among South Asian (aOR 0.84, 95% CI [0.80, 0.87], $p < 0.001$) and black (aOR 0.88, 95% CI [0.83, 0.94], $p < 0.001$) individuals, but increased the adjusted odds in other subgroups including older adults (aged 60 to 79 years: aOR 1.11, 95% CI [1.09, 1.12], $p < 0.001$) and the least deprived IMD quintile (aOR 1.17, 95% CI [1.15, 1.20], $p < 0.001$) (S4–S9 Tables).

## Discussion

The 3 years since the onset of the COVID-19 pandemic saw profound societal change in how individuals live, work, and interact with each other. However, the impact of these changes on patterns of weight gain have not been well characterised. We are, to our knowledge, the first to use a national health database [21] to describe patterns of weight gain during the pandemic. We report, to our knowledge, novel findings from our analyses of BMI data in the EHRs of over 17 million adults living in England. At population level, there was a modest increase in the median BMI between the prepandemic year and the year beginning March 2021. At

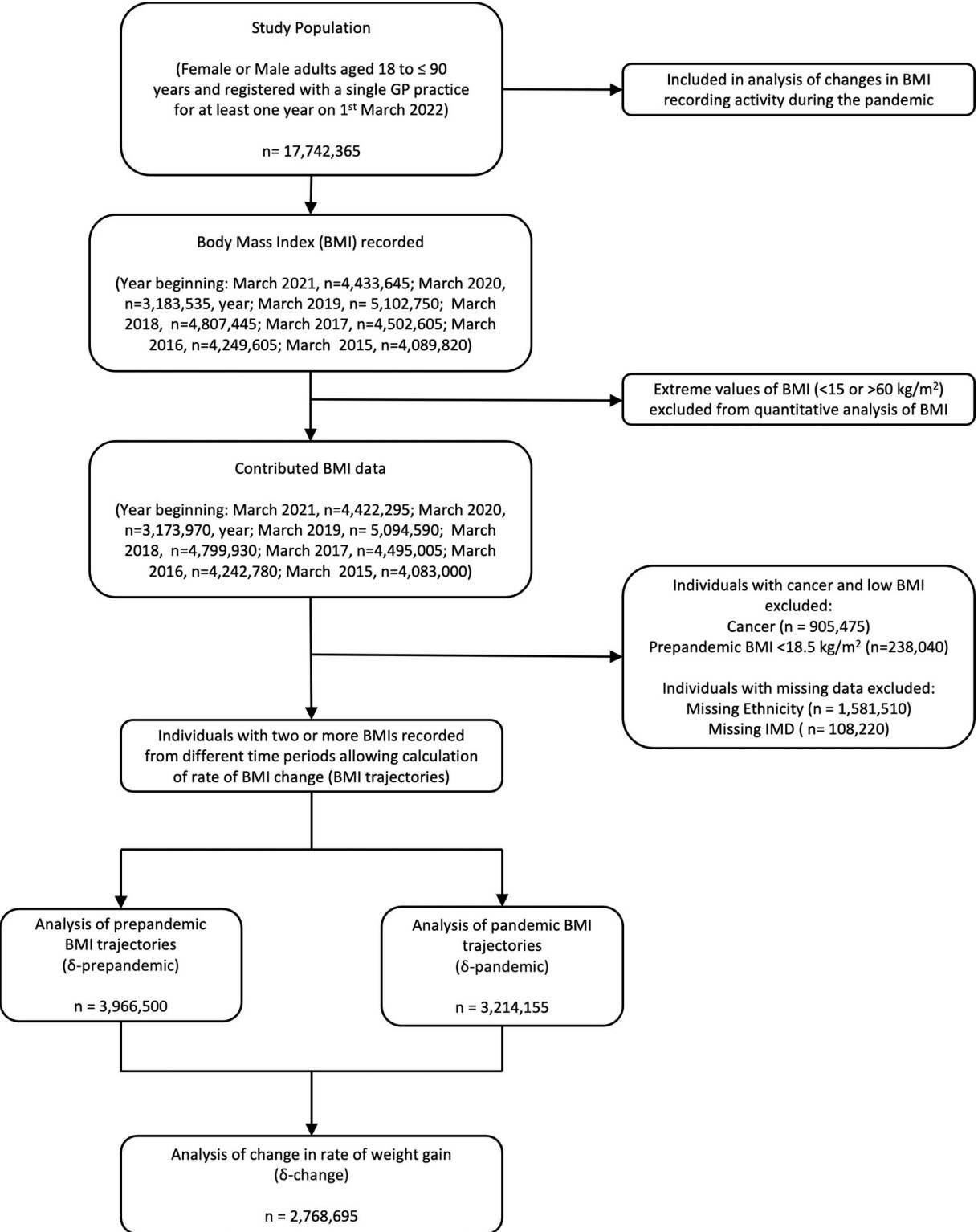

**Fig 1. Flowchart for selection of the study population.** GP, general practitioner; kg, kilogram; m, metre; IMD, index of multiple deprivation.

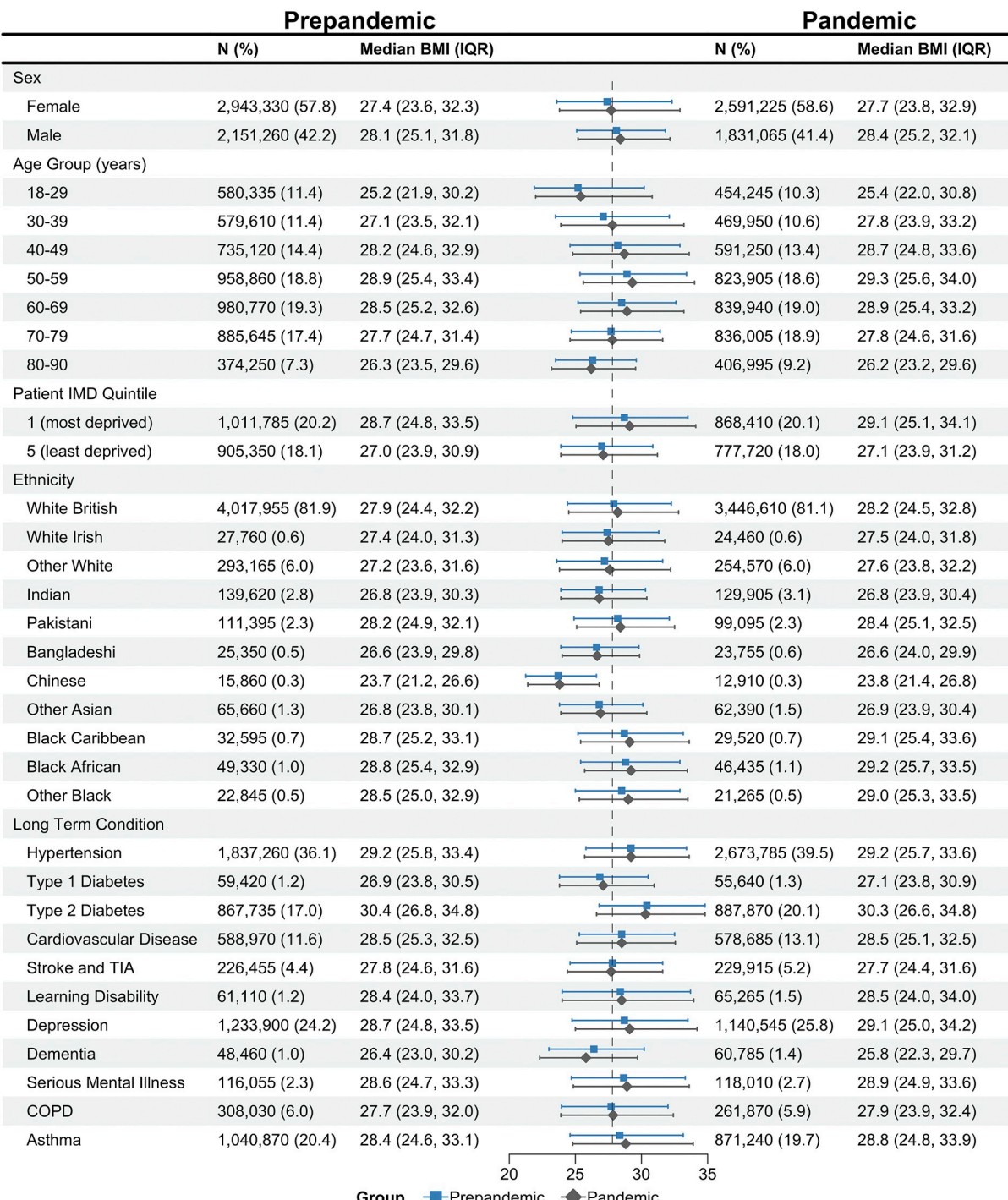

| | Prepandemic | | | Pandemic | |
|---|---|---|---|---|---|
| | N (%) | Median BMI (IQR) | | N (%) | Median BMI (IQR) |
| **Sex** | | | | | |
| Female | 2,943,330 (57.8) | 27.4 (23.6, 32.3) | | 2,591,225 (58.6) | 27.7 (23.8, 32.9) |
| Male | 2,151,260 (42.2) | 28.1 (25.1, 31.8) | | 1,831,065 (41.4) | 28.4 (25.2, 32.1) |
| **Age Group (years)** | | | | | |
| 18-29 | 580,335 (11.4) | 25.2 (21.9, 30.2) | | 454,245 (10.3) | 25.4 (22.0, 30.8) |
| 30-39 | 579,610 (11.4) | 27.1 (23.5, 32.1) | | 469,950 (10.6) | 27.8 (23.9, 33.2) |
| 40-49 | 735,120 (14.4) | 28.2 (24.6, 32.9) | | 591,250 (13.4) | 28.7 (24.8, 33.6) |
| 50-59 | 958,860 (18.8) | 28.9 (25.4, 33.4) | | 823,905 (18.6) | 29.3 (25.6, 34.0) |
| 60-69 | 980,770 (19.3) | 28.5 (25.2, 32.6) | | 839,940 (19.0) | 28.9 (25.4, 33.2) |
| 70-79 | 885,645 (17.4) | 27.7 (24.7, 31.4) | | 836,005 (18.9) | 27.8 (24.6, 31.6) |
| 80-90 | 374,250 (7.3) | 26.3 (23.5, 29.6) | | 406,995 (9.2) | 26.2 (23.2, 29.6) |
| **Patient IMD Quintile** | | | | | |
| 1 (most deprived) | 1,011,785 (20.2) | 28.7 (24.8, 33.5) | | 868,410 (20.1) | 29.1 (25.1, 34.1) |
| 5 (least deprived) | 905,350 (18.1) | 27.0 (23.9, 30.9) | | 777,720 (18.0) | 27.1 (23.9, 31.2) |
| **Ethnicity** | | | | | |
| White British | 4,017,955 (81.9) | 27.9 (24.4, 32.2) | | 3,446,610 (81.1) | 28.2 (24.5, 32.8) |
| White Irish | 27,760 (0.6) | 27.4 (24.0, 31.3) | | 24,460 (0.6) | 27.5 (24.0, 31.8) |
| Other White | 293,165 (6.0) | 27.2 (23.6, 31.6) | | 254,570 (6.0) | 27.6 (23.8, 32.2) |
| Indian | 139,620 (2.8) | 26.8 (23.9, 30.3) | | 129,905 (3.1) | 26.8 (23.9, 30.4) |
| Pakistani | 111,395 (2.3) | 28.2 (24.9, 32.1) | | 99,095 (2.3) | 28.4 (25.1, 32.5) |
| Bangladeshi | 25,350 (0.5) | 26.6 (23.9, 29.8) | | 23,755 (0.6) | 26.6 (24.0, 29.9) |
| Chinese | 15,860 (0.3) | 23.7 (21.2, 26.6) | | 12,910 (0.3) | 23.8 (21.4, 26.8) |
| Other Asian | 65,660 (1.3) | 26.8 (23.8, 30.1) | | 62,390 (1.5) | 26.9 (23.9, 30.4) |
| Black Caribbean | 32,595 (0.7) | 28.7 (25.2, 33.1) | | 29,520 (0.7) | 29.1 (25.4, 33.6) |
| Black African | 49,330 (1.0) | 28.8 (25.4, 32.9) | | 46,435 (1.1) | 29.2 (25.7, 33.5) |
| Other Black | 22,845 (0.5) | 28.5 (25.0, 32.9) | | 21,265 (0.5) | 29.0 (25.3, 33.5) |
| **Long Term Condition** | | | | | |
| Hypertension | 1,837,260 (36.1) | 29.2 (25.8, 33.4) | | 2,673,785 (39.5) | 29.2 (25.7, 33.6) |
| Type 1 Diabetes | 59,420 (1.2) | 26.9 (23.8, 30.5) | | 55,640 (1.3) | 27.1 (23.8, 30.9) |
| Type 2 Diabetes | 867,735 (17.0) | 30.4 (26.8, 34.8) | | 887,870 (20.1) | 30.3 (26.6, 34.8) |
| Cardiovascular Disease | 588,970 (11.6) | 28.5 (25.3, 32.5) | | 578,685 (13.1) | 28.5 (25.1, 32.5) |
| Stroke and TIA | 226,455 (4.4) | 27.8 (24.6, 31.6) | | 229,915 (5.2) | 27.7 (24.4, 31.6) |
| Learning Disability | 61,110 (1.2) | 28.4 (24.0, 33.7) | | 65,265 (1.5) | 28.5 (24.0, 34.0) |
| Depression | 1,233,900 (24.2) | 28.7 (24.8, 33.5) | | 1,140,545 (25.8) | 29.1 (25.0, 34.2) |
| Dementia | 48,460 (1.0) | 26.4 (23.0, 30.2) | | 60,785 (1.4) | 25.8 (22.3, 29.7) |
| Serious Mental Illness | 116,055 (2.3) | 28.6 (24.7, 33.3) | | 118,010 (2.7) | 28.9 (24.9, 33.6) |
| COPD | 308,030 (6.0) | 27.7 (23.9, 32.0) | | 261,870 (5.9) | 27.9 (23.9, 32.4) |
| Asthma | 1,040,870 (20.4) | 28.4 (24.6, 33.1) | | 871,240 (19.7) | 28.8 (24.8, 33.9) |

**Group** ■ Prepandemic ◆ Pandemic

**Fig 2. Median recorded BMI of the study population in the prepandemic and pandemic period. Results are presented for the prepandemic period based on BMI data (weight in kilograms/height in metres squared) recorded in the year beginning March 2019, and for the pandemic period based on BMI data recorded in the year beginning March 2021.** *N* (%): Number (and percentage) of adults contributing data to the analyses from each population subgroup. IQR, interquartile range; IMD, index of multiple deprivation; SMI, serious mental illness (includes bipolar disorder and psychosis); TIA, transient ischaemic attack; COPD, chronic obstructive pulmonary disease; BMI, body mass index. Dot and whiskers plot: dot represents median BMI, whiskers show IQR of BMI. Data for individuals of mixed ethnicity can be found in the accompanying table.

| | Prepandemic | | | | Pandemic | | |
|---|---|---|---|---|---|---|---|
| | **N** | **Rapid (%)** | **aOR (95% CI)** | | **N** | **Rapid (%)** | **aOR (95% CI)** |
| **Sex** | | | | | | | |
| Female | 2,381,345 | 30.11 | 1 | | 1,898,510 | 32.87 | 1 |
| Male | 1,585,155 | 22.13 | 0.80 (0.79,0.80) | | 1,315,645 | 23.90 | 0.76 (0.76,0.76) |
| **Age Group (years)** | | | | | | | |
| 18-29 | 274,540 | 46.00 | 1 | | 267,605 | 44.18 | 1 |
| 30-39 | 466,285 | 38.70 | 0.76 (0.75,0.76) | | 341,870 | 40.78 | 0.90 (0.89,0.91) |
| 40-49 | 545,760 | 32.28 | 0.60 (0.59,0.60) | | 415,805 | 34.58 | 0.73 (0.72,0.74) |
| 50-59 | 757,830 | 27.68 | 0.49 (0.48,0.49) | | 603,670 | 30.08 | 0.60 (0.60,0.61) |
| 60-69 | 771,890 | 22.68 | 0.38 (0.37,0.38) | | 642,120 | 25.54 | 0.49 (0.48,0.49) |
| 70-79 | 757,405 | 18.43 | 0.29 (0.29,0.29) | | 635,175 | 21.28 | 0.38 (0.38,0.39) |
| 80-90 | 392,785 | 15.41 | 0.23 (0.23,0.23) | | 307,910 | 18.26 | 0.31 (0.31,0.31) |
| **Patient IMD Quintile** | | | | | | | |
| 1 (Most Deprived) | 836,480 | 29.86 | 1 | | 686,100 | 32.61 | 1 |
| 5 (Least Deprived) | 692,065 | 24.39 | 0.82 (0.82,0.83) | | 549,495 | 26.07 | 0.77 (0.77,0.78) |
| **Ethnicity** | | | | | | | |
| White British | 3,275,985 | 26.94 | 1 | | 2,631,235 | 29.42 | 1 |
| White Irish | 21,500 | 24.14 | 0.97 (0.94,1.00) | | 17,590 | 26.46 | 0.94 (0.91,0.97) |
| Other White | 211,145 | 29.17 | 0.98 (0.97,0.99) | | 172,260 | 30.99 | 0.94 (0.93,0.95) |
| Indian | 115,585 | 22.63 | 0.74 (0.73,0.76) | | 99,750 | 23.12 | 0.68 (0.67,0.69) |
| Pakistani | 99,045 | 27.04 | 0.79 (0.78,0.80) | | 80,400 | 27.56 | 0.72 (0.71,0.74) |
| Bangladeshi | 21,230 | 25.15 | 0.70 (0.68,0.72) | | 18,965 | 24.65 | 0.60 (0.58,0.62) |
| Chinese | 10,185 | 19.88 | 0.62 (0.59,0.66) | | 8,245 | 20.13 | 0.56 (0.53,0.59) |
| Other Asian | 51,865 | 24.70 | 0.77 (0.75,0.78) | | 46,590 | 25.35 | 0.70 (0.69,0.72) |
| Black Caribbean | 37,745 | 29.29 | 0.85 (0.83,0.87) | | 32,875 | 32.41 | 0.91 (0.89,0.94) |
| Black African | 27,170 | 24.60 | 0.89 (0.87,0.91) | | 23,050 | 28.79 | 0.91 (0.89,0.93) |
| Other Black | 17,295 | 28.45 | 0.89 (0.86,0.92) | | 15,020 | 31.89 | 0.92 (0.89,0.95) |
| **Long Term Condition** | | | | | | | |
| Hypertension | 1,637,165 | 21.85 | 1.04 (1.03,1.04) | | 1,436,070 | 24.74 | 1.03 (1.03,1.04) |
| Type 1 Diabetes | 55,810 | 27.73 | 0.93 (0.91,0.94) | | 51,250 | 32.57 | 1.06 (1.04,1.08) |
| Type 2 Diabetes | 820,390 | 19.25 | 0.79 (0.79,0.80) | | 793,770 | 20.73 | 0.71 (0.71,0.72) |
| Cardiovascular Disease | 560,030 | 20.87 | 1.07 (1.06,1.08) | | 487,915 | 23.73 | 1.07 (1.06,1.08) |
| Stroke and TIA | 219,360 | 20.87 | 1.06 (1.05,1.08) | | 187,580 | 23.72 | 1.05 (1.04,1.07) |
| Learning Difficulties | 54,580 | 34.00 | 1.08 (1.07,1.10) | | 55,750 | 36.34 | 1.10 (1.08,1.12) |
| Depression | 1,105,540 | 30.90 | 1.19 (1.18,1.19) | | 887,240 | 33.36 | 1.18 (1.17,1.18) |
| Serious Mental Illness | 100,025 | 31.87 | 1.20 (1.18,1.22) | | 101,395 | 35.16 | 1.24 (1.22,1.27) |
| Dementia | 58,670 | 20.76 | 1.20 (1.18,1.23) | | 47,225 | 24.69 | 1.23 (1.21,1.24) |
| Asthma | 939,925 | 28.88 | 1.03 (1.03,1.04) | | 709,730 | 31.32 | 1.03 (1.03,1.04) |
| COPD | 299,095 | 24.44 | 1.22 (1.21,1.23) | | 227,935 | 26.19 | 1.12 (1.10,1.13) |

0.4  0.7  1  1.2

**Group** ■ Prepandemic ◆ Pandemic

**Fig 3. Estimated associations between sociodemographic and clinical characteristics and odds of rapid weight gain (>0.5 kg/m$^2$/year) during the prepandemic and pandemic periods among adults living in England during the COVID-19 pandemic.** Rate of weight gain calculated from data recorded in the routine healthcare record. kg, kilograms; m, metre; *N*, number of adults contributing data to the analyses. Rapid (%): Percentage of adults who gained weight rapidly. aOR, adjusted odds ratio of rapid weight gain adjusted for age and sex. aOR for LTCs presented in comparison to the baseline group without those conditions recorded. CI, confidence interval; IMD, index of multiple deprivation; SMI, serious mental illness (includes bipolar disorder

and psychosis); COPD, chronic obstructive pulmonary disease; TIA, transient ischaemic attack. Dot and whiskers plot: dot represents aOR estimate, whiskers show 95% CI of aOR. Data for individuals of mixed ethnicity can be found in the accompanying table. COVID-19, Coronavirus Disease 2019; LTC, long-term condition.

individual level, we showed that, among adults with BMI values recorded in the EHR, women, younger adults (18 to 29 years), and those living in the most socioeconomically deprived areas had the greatest estimated odds of unhealthy patterns of weight gain, namely rapid weight gain during the pandemic and an extreme acceleration in rate of weight gain between the prepandemic and pandemic periods. Individuals with a history of LTCs also had greater odds of unhealthy weight gain than those without LTCs, and mental health conditions, including depression, SMI, and learning difficulties, were associated with greater odds than physical health conditions such as hypertension.

Previous research in this area has been limited to small voluntary surveys or routine health data from populations with specific comorbidities or in geographical regions where access to healthcare is stratified by structures of payment and insurance [10–13,22]. Registration with a GP is almost universal in England and all healthcare activity should be reported back to primary care. Therefore, routinely collected EHRs can provide a national database of health data [15] available during the pandemic. We have used a large-scale and contemporaneous EHR dataset to robustly reproduce prior observations that women [11], young adults [11], those living in deprivation [12], and those with LTCs [12,13] may be the most affected by unhealthy patterns of pandemic weight gain and demonstrate this at population scale. Our findings suggest the increased risk of weight gain seen among young adults prepandemic [14] was exacerbated during the pandemic. The increased risk seen among women may reflect gender disparities in the impact of the pandemic on social factors such as employment loss and caring responsibilities, with a consequent influence on health behaviours [23]. The increased risk among those living in areas of greater deprivation reflects prepandemic trends and is likely to have multifactorial aetiology including food poverty, reduced opportunity for physical activity, and an increased burden of physical and mental health conditions [24]. In our analysis, instead of controlling for LTCs as prior studies have done [1,14], we describe the associations between patterns of weight gain and different clinical and mental health characteristics. This approach uncovered important inequality between people with prior mental health versus physical health conditions, with the former having a greater risk of unhealthy patterns of weight gain. These differences may reflect associations between disordered eating, reduced physical activity, and poor mental health exacerbated by the pandemic [25,26]. As a descriptive study we cannot assign a causal relationship between mental health conditions and unhealthy patterns of weight gain, nor comment on pathways that may mediate apparent associations. However, our findings highlight the need for parity of esteem [27] between physical and mental health conditions in research and when prioritising groups to be supported by weight loss interventions.

The key strengths of this study are the quality, scale and representativeness of the health record data used [21]. OpenSAFELY-TPP provides near real-time access to the primary care records of roughly 40% of the population [21]. In England, registration with a National Health Service (NHS) GP is almost universal and primary care hosts and records data from 90% of all patient consultations in the NHS, with particular responsibility for preventative care and routine management of LTCs [28]. Therefore, our study has been able to make robust observations that have immediate relevance to the delivery of health care in the post-pandemic period. These findings could be rapidly implemented in the routine clinical care that has generated them, for example, through the practice reorganisation or redistribution of care to target those

| | N | Extreme Acceleration (%) | | aOR (95% CI) |
|---|---|---|---|---|
| **Sex** | | | | |
| Female | 1,612,850 | 11.58 | | 1 |
| Male | 1,155,845 | 7.79 | | 0.72 (0.72, 0.73) |
| **Age Group (years)** | | | | |
| 18-29 | 161,655 | 14.86 | | 1 |
| 30-39 | 253,415 | 14.46 | | 1.00 (0.98, 1.02) |
| 40-49 | 310,045 | 11.85 | | 0.85 (0.84, 0.87) |
| 50-59 | 479,655 | 10.79 | | 0.79 (0.78, 0.80) |
| 60-69 | 573,030 | 9.28 | | 0.68 (0.67, 0.69) |
| 70-79 | 649,920 | 7.68 | | 0.55 (0.54, 0.56) |
| 80-90 | 340,975 | 7.21 | | 0.51 (0.50, 0.52) |
| **Patient IMD Quintile** | | | | |
| 1 (most deprived) | 592,805 | 11.67 | | 1 |
| 5 (least deprived) | 472,800 | 8.46 | | 0.72 (0.71, 0.73) |
| **Ethnicity** | | | | |
| White British | 2,306,865 | 10.20 | | 1 |
| White Irish | 15,475 | 9.40 | | 0.98 (0.93, 1.03) |
| Other White | 131,960 | 10.23 | | 0.92 (0.90, 0.94) |
| Indian | 82,245 | 7.05 | | 0.63 (0.62, 0.65) |
| Pakistani | 68,000 | 8.33 | | 0.65 (0.63, 0.67) |
| Bangladeshi | 15,025 | 7.22 | | 0.55 (0.51, 0.58) |
| Chinese | 6,395 | 5.24 | | 0.47 (0.42, 0.52) |
| Other Asian | 36,420 | 7.34 | | 0.62 (0.60, 0.65) |
| Black Caribbean | 20,330 | 10.28 | | 0.93 (0.89, 0.97) |
| Black African | 23,970 | 10.43 | | 0.84 (0.81, 0.88) |
| Other Black | 11,300 | 11.15 | | 0.93 (0.87, 0.98) |
| **Long Term Condition** | | | | |
| Hypertension | 1,399,300 | 8.91 | | 1.07 (1.06, 1.08) |
| Type 1 Diabetes | 47,910 | 9.38 | | 0.87 (0.85, 0.90) |
| Type 2 Diabetes | 805,315 | 9.06 | | 1.07 (1.06, 1.08) |
| Cardiovascular Disease | 498,305 | 8.66 | | 1.10 (1.08, 1.11) |
| Stroke and TIA | 190,500 | 9.05 | | 1.12 (1.10, 1.14) |
| Learning Difficulties | 49,865 | 13.89 | | 1.25 (1.21, 1.28) |
| Depression | 777,355 | 12.49 | | 1.28 (1.27, 1.29) |
| Serious Mental Illness | 88,375 | 14.66 | | 1.47 (1.44, 1.50) |
| Dementia | 48,415 | 12.27 | | 1.68 (1.64, 1.73) |
| Asthma | 638,485 | 11.23 | | 1.09 (1.08, 1.10) |
| COPD | 245,510 | 9.84 | | 1.14 (1.12, 1.16) |

0.6 0.8 1 1.2 1.4 1.6

← Reduced Odds    Increased Odds →

**Fig 4. Estimated associations between sociodemographic and clinical characteristics and extreme acceleration in rate of weight gain between the prepandemic and pandemic periods among adults living in England during the COVID-19 pandemic.** Extreme acceleration in rate of weight gain is defined as δ-change $\geq$1.84 kg/m$^2$/year. δ-change refers to the change in rate of weight gain (δ) in kg/m$^2$/year between the prepandemic (δ-prepandemic) and pandemic (δ-pandemic) periods (δ-change = δ-pandemic—δ-prepandemic). *N*: Number of adults contributing data to the analyses. Extreme

acceleration (%): Percentage of adults who had an extreme acceleration in rate of weight gain. aOR, adjusted odds ratio of extreme acceleration in rate of weight gain adjusted for age and sex. aOR for LTCs presented in comparison to the baseline group without those conditions recorded. CI, confidence interval; IMD, index of multiple deprivation; SMI, serious mental illness (includes bipolar disorder and psychosis); COPD, chronic obstructive pulmonary disease; TIA, transient ischaemic attack. Dot and whiskers plot: dot represents aOR estimate, whiskers show 95% CI of aOR. Data for individuals of mixed ethnicity can be found in the accompanying table. COVID-19, Coronavirus Disease 2019; LTC, long-term condition.

at most need. We are, to our knowledge, the first study to present detailed analyses of associations between a wide range of sociodemographic and clinical characteristics and patterns of weight gain. Due to the size of the dataset we had to limit the analysis to common conditions which have established care pathways associated with national quality frameworks. We would welcome further research into how the pandemic influenced patterns of weight gain among adults living with other clinical conditions that could impact rates of weight gain, such as eating disorders.

We recognise the limitations of our work. We categorised age for ease of interpretation and application to clinical practice. But this may have flattened effects within categories and over emphasised the step change in effect between categories. We restricted our analyses to individuals registered with a GP practice for at least 1 year to minimise the impact of incomplete data. However, this may have introduced survival bias, e.g., if individuals with the unhealthiest patterns of weight gain were more likely to die during the pandemic. BMI is recorded in EHRs when clinically indicated, this may be systematically as part of the routine monitoring of LTCs such as T2D or opportunistically when relevant to the clinical consultation. Despite this risk of information bias, prepandemic EHRs produced BMI trajectory estimates comparable to representative national surveys [15]. However, we report changes in BMI recording activity that may have introduced new bias, with a decline in BMI recording after the onset of the pandemic which had only partly recovered by the year beginning March 2021, reflecting wider patterns of primary care activity during the pandemic [29]. Notably, the greatest recovery in BMI recording was among individuals with LTCs requiring annual BMI recording, e.g., T2D. This leads to a higher prevalence of LTCs among individuals contributing pandemic BMI data (Table 1) and the possibility that the protective effect of T2D against weight gain is in fact due to collider bias [30]. Alternatively, this may be a true effect arising from the awareness that COVID-19 infection confers greater risk to people with obesity and LTCs such as T2D [5], leading to more proactive and intensive weight management in primary care, or adoption of healthy behaviours. The observed protective effect of T1D against extreme acceleration may also reflect bias; however, there was early evidence of improved blood glucose control among people living with T1D during the pandemic [31]. However a deceleration in rate of weight gain, or weight loss, can also reflect poorly glycaemic control among people with diabetes during the pandemic, as has been reported in other studies [32]. Further research would be required to unpick how diabetes impacted weight and glycaemic control during the pandemic. Pandemic-associated changes in clinical activity are likely to have introduced biases that are harder to characterise, such as an increase in the number of self-reported weight or height measures entered into clinical record during remote consultations [33]. This could cause systematic bias if some sociodemographic or clinical groups were more likely to have self-reported values recorded than others during the pandemic. As there is no clinical code to indicate whether weight and height measures are self-reported, the impact of this bias cannot be assessed using the available data. Further research, including representative population health surveys, would be required to investigate this further.

Another limitation relates to missing data. We have undertaken a complete case analysis, removing individuals with missing ethnicity and IMD data. This approach precludes

generalisability of findings to those missing data in these fields, e.g., those missing IMD data due to unstable accommodation, but gives robust evidence in everybody else. Multiple imputation models would not overcome this limitation, as the Missing at Random assumption is unlikely to hold [34]. Some prepandemic analyses of BMI trajectories have used imputation, adjusting estimates with population-survey data to account for the Missing Not at Random assumption [14]. We could not replicate this approach due to the suspension of such surveys during the pandemic.

As a descriptive study, we did not seek to assign a causal relationship between the exposures and the outcomes, nor identify the pathways that mediate observed associations. Therefore, the observed relationship between exposures, such as mental health conditions, and unhealthy patterns of weight gain may have been mediated through pathways including comorbid physical health conditions. Our findings emphasise the need for further research to unpick the complex relationships between physical health, mental health, sociodemographic characteristics, and patterns of weight gain.

In conclusion, our study is, to our knowledge, the largest and most representative analysis of weight changes associated with the pandemic to date. We identify that women, young adults, those living in the most deprived areas, as well as those with mental health conditions were at increased risk of unhealthy patterns of weight gain during the pandemic. We demonstrate the value of routinely collected data to understand the impact of a pandemic at population scale and identify health inequalities and areas of focus. Rapid weight gain is an independent risk factor for cardiovascular disease and mortality [1] and associated with an increased risk of progressions to obesity [14]. Overweight and obesity is one of the greatest global public health challenges of our time and projected to cost the global economy 3.3% of gross domestic product by 2060. In response, the World Health Organisation acceleration plan to stop obesity emphasises the need for country-level interventions such as fiscal measures to reduce consumption of ultra-processed food and beverages, and health services for the preventions and management of obesity [35]. Our findings identified clear targets for such preventative interventions. We would welcome further focussed research to understand the complex causal pathways that underlie the associations we describe in this analysis. Future pandemic planning should consider how to mitigate the unequal indirect impact of the pandemic on cardiometabolic risk factors such as rate of weight gain to prevent preexisting health inequalities from becoming further exacerbated.

## Supporting information

**S1 Checklist. REporting of studies Conducted using Observational Routinely-collected health Data (RECORD) Statement Checklist.**
(DOCX)

**S1 Protocol. Prospective study protocol submitted to the London School of Hygiene and Tropical Medicine Ethics Boards.**
(DOCX)

**S1 Appendix. Information Governance Statement.**
(DOCX)

**S2 Appendix. Calculation of δ-prepandemic, δ-pandemic and δ-change.**
(DOCX)

**S3 Appendix. Defining a population of extreme accelerators (distribution of δ-change).**
(DOCX)

**S1 Table. Average body mass index (BMI) recorded in the routine healthcare records of adults living in England before and after the onset of the pandemic.**
(DOCX)

**S2 Table. Proportion of adults living in England with a BMI value recorded in their routine health care record in the period before and after the onset of the COVID-19 pandemic.**
(DOCX)

**S3 Table. Rate of weight gain of adults living in England before and after the onset of the COVID-19 pandemic calculated from measures of weight recorded in the routine healthcare record.**
(DOCX)

**S4 Table. Estimated associations between sociodemographic and clinical characteristics and odds of rapid weight gain during the prepandemic and pandemic period.**
(DOCX)

**S5 Table. Estimated associations between sociodemographic and clinical characteristics and odds of extreme acceleration in rate of weight gain between the prepandemic and pandemic period.**
(DOCX)

**S6 Table. Associations between sociodemographic and clinical characteristics and odds of extreme acceleration in rate of weight gain during the pandemic in analyses stratified by age.**
(DOCX)

**S7 Table. Associations between sociodemographic and clinical characteristics and odds of extreme acceleration in rate of weight gain during the pandemic in analyses stratified by sex.**
(DOCX)

**S8 Table. Associations between sociodemographic and clinical characteristics and odds of extreme acceleration in rate of weight gain during the pandemic in analyses stratified by ethnicity.**
(DOCX)

**S9 Table. Associations between sociodemographic and clinical characteristics and odds of extreme acceleration in rate of weight gain during the pandemic in analyses stratified by index of multiple deprivation (IMD).**
(DOCX)

## Acknowledgments

We are very grateful for all the support received from the TPP Technical Operations team throughout this work, and for generous assistance from the information governance and database teams at NHS England and the NHS England Transformation Directorate. The OpenSAFELY collaborative is composed of Alex J Walker, Amelia CA Green, Amir Mehrkar, Andrea L Schaffer, Andrew D Brown, Ben Goldacre, Ben FC Butler-Cole, Brian MacKenna, Caroline E Morton, Caroline E Walters, Catherine L Stables, Christine Cunningham, Christopher Bates, Christopher Wood, Colm D Andrews, David Evans, Frank Hester, George Hickman, Helen J Curtis, Henry Drysdale, Iain Dillingham, Jessica Morley, Jon Massey, Jonathan Cockburn, John Parry, Liam C Hart, Linda Nab, Lisa EM Hopcroft, Louis Fisher, Lucy Bridges,

Milan Wiedemann, Nicholas J DeVito, Orla Macdonald, Peter Inglesby, Pete Stokes, Rebecca M Smith, Richard Croker, Robin Y Park, Rose Higgins, Sam Harper, Sebastian CJ Bacon, Simon Davy, Steven Maude, Thomas O'Dwyer, Tom Ward, Victoria Speed, and William J Hulme.

## Author Contributions

**Conceptualization:** Miriam Samuel, Robin Y. Park, Fabiola Eto, Caroline E. Morton, Kamlesh Khunti, Rohini Mathur, Jonathan Valabhji, Brian MacKenna, Sarah Finer.

**Data curation:** Miriam Samuel, Robin Y. Park, Caroline E. Morton, Sebastian Bacon, Amir Mehrkar, Iain Dillingham, Peter Inglesby, William J. Hulme, Brian MacKenna.

**Formal analysis:** Miriam Samuel, Robin Y. Park.

**Funding acquisition:** Miriam Samuel, Sebastian Bacon, Amir Mehrkar.

**Investigation:** Miriam Samuel.

**Methodology:** Miriam Samuel, Robin Y. Park, Sophie V. Eastwood, Fabiola Eto, Caroline E. Morton, Daniel Stow, Rohini Mathur, Brian MacKenna, Sarah Finer.

**Project administration:** Miriam Samuel, Caroline E. Morton, Sebastian Bacon, Amir Mehrkar, Jessica Morley, Iain Dillingham, Peter Inglesby, William J. Hulme, Brian MacKenna.

**Resources:** Miriam Samuel, Caroline E. Morton, Sebastian Bacon, Amir Mehrkar, Jessica Morley, Iain Dillingham, Peter Inglesby, William J. Hulme, Brian MacKenna, Sarah Finer.

**Software:** Miriam Samuel, Peter Inglesby, William J. Hulme, Brian MacKenna.

**Supervision:** Robin Y. Park, Sophie V. Eastwood, Fabiola Eto, Caroline E. Morton, Daniel Stow, William J. Hulme, Kamlesh Khunti, Rohini Mathur, Jonathan Valabhji, Brian MacKenna, Sarah Finer.

**Validation:** Miriam Samuel.

**Visualization:** Miriam Samuel.

**Writing – original draft:** Miriam Samuel.

**Writing – review & editing:** Miriam Samuel, Robin Y. Park, Sophie V. Eastwood, Fabiola Eto, Caroline E. Morton, Daniel Stow, Sebastian Bacon, Amir Mehrkar, Jessica Morley, Iain Dillingham, Peter Inglesby, William J. Hulme, Kamlesh Khunti, Rohini Mathur, Jonathan Valabhji, Brian MacKenna, Sarah Finer.

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
