## [Editor Report · Decision Letter 0]

1 Jun 2023

Dear Dr Samuel, 

Thank you for submitting your manuscript entitled "Trends in weight gain recorded in English primary care before and during the Coronavirus-19 pandemic: an observational cohort study using the OpenSAFELY platform" for consideration by PLOS Medicine.

Your manuscript has now been evaluated by the PLOS Medicine editorial staff as well as by an academic editor with relevant expertise and I am writing to let you know that we would like to send your submission out for external peer review.

Please re-submit your manuscript within two working days, i.e. by Jun 05 2023 11:59PM.

Kind regards,

Alexandra Schaefer, PhD

Associate Editor

PLOS Medicine

---

## [Decision Letter · Decision Letter 1]

24 Aug 2023

Dear Dr. Samuel,

Thank you very much for submitting your manuscript "Trends in weight gain recorded in English primary care before and during the Coronavirus-19 pandemic: an observational cohort study using the OpenSAFELY platform" (PMEDICINE-D-23-01514R1) for consideration at PLOS Medicine. 

Your paper was evaluated by an associate editor and discussed among all the editors here. It was also discussed with an academic editor with relevant expertise, and sent to independent reviewers, including a statistical reviewer. The reviews are appended at the bottom of this email and any accompanying reviewer attachments can be seen via the link below:

[LINK]

In light of these reviews, I am afraid that we will not be able to accept the manuscript for publication in the journal in its current form, but we would like to consider a revised version that addresses the reviewers' and editors' comments. Obviously we cannot make any decision about publication until we have seen the revised manuscript and your response, and we plan to seek re-review by one or more of the reviewers. 

We expect to receive your revised manuscript by Sep 14 2023 11:59PM. Please email us (plosmedicine@plos.org) if you have any questions or concerns.

We look forward to receiving your revised manuscript. 

Sincerely,

Alexandra Schaefer, PhD

PLOS Medicine

plosmedicine.org

GENERAL COMMENTS

Please respond to all editor and reviewer comments.

Please include page numbers and line numbers in the manuscript file. Use continuous line numbers (do not restart the numbering on each page).

Please cite the reference numbers in square brackets (e.g., “We used the techniques developed by our colleagues [19] to analyze the data”). Citations should be preceding punctuation.

Please cite your Supporting Information as outlined here: https://journals.plos.org/plosmedicine/s/supporting-information

Please remove the sections ‘Role of Funding Source’, ‘Declaration of Interests’, ‘Data Sharing’ from the main manuscript. This information should only be included in the according section in the online submission form.

Please ensure consistency in your number format and revise your manuscript accordingly including the Supplementary Material (51·4% or 51.4%; 1000 or 1,000).

We have noticed that you mostly define the time periods examined in your study as "pandemic" and "pre-pandemic", whereas in some figure/table titles you refer to the time periods as "before and after the onset of the COVID-19 pandemic". We suggest that you adopt a consistent format for clarity for the reader.

Please comment on the definition of the age groups studied in your study. You have stated that "data were extracted on all adults aged ≥18 to ≤90 years [...]" but then define the oldest age group as "80-89". In some of the supplementary tables, '90' appears to be a single age group, but not, for example, in Table 1, which shows the baseline characteristics of the study population.

ACADEMIC EDITOR COMMENTS

While the authors state that "this is a descriptive study," one could argue that there is a causal tinge to the analysis. Some of the comments by Reviewer 2 and 3 are getting to this point in the questioning of covariates and requesting to include baseline BMI in the models.

Would the paper be better framed descriptively by prioritizing simpler unadjusted or age-adjusted or or stratified models, rather than fully multivariable models? What do the fully adjusted models add? For example, what are readers to make of models that adjust for both ethnicity and deprivation? This might diminish the extend of disparities that are observed from a descriptive perspective.

It seems hard to interpret each individual odds ratio in any way that is not casually influenced and susceptible to the Table 2 fallacy.

EDITORIAL COMMENTS

In your discussion, you refer to post-pandemic research, policies and interventions targeting BMI, and we suggest that you expand your discussion to include future pre-pandemic planning and preventive interventions.

FINANCIAL DISCLOSURE

The funding statement should include: specific grant numbers, initials of authors who received each award, URLs to sponsors’ websites. 

ABSTRACT

Please structure your abstract using the PLOS Medicine headings (Background, Methods and Findings, Conclusions). Please combine the Methods and Findings sections into one section, “Methods and findings”.

PLOS Medicine requests that main results are quantified with 95% CIs as well as p values. When reporting p values please report as p<0.001 and where higher as the exact p value p=0.002, for example. For the purposes of transparent data reporting, if not including the aforementioned please clearly state the reasons why not.

Please include any important dependent variables that are adjusted for in the analyses.

Throughout, suggest reporting statistical information as follows to improve clarity for the reader “22% (95% CI [13%,28%]; p</=)”. Please amend throughout the abstract and main manuscript.

Please note the use of commas to separate upper and lower bounds, as opposed to hyphens as these can be confused with reporting of negative values.

Please define all abbreviations used for statistical reporting at first use.

When a p value is given, please specify the statistical test used to determine it.

Please define 'NHS‘, ‚IQR‘, ‚aOR’, ‘IMD’.

Abstract Background: Provide the context of why the study is important. The final sentence should clearly state the study question.

Abstract Methods and Findings:

* Please ensure that all numbers presented in the abstract are present and identical to numbers presented in the main manuscript text.

* Please include the population (age range should be defined for ‘adults’) and setting.

Abstract Conclusions:

* Please avoid vague statements such as "these results have major implications for policy/clinical care". Mention only specific implications substantiated by the results.

AUTHOR SUMMARY

At this stage, we ask that you include a short, non-technical Author Summary of your research to make findings accessible to a wide audience that includes both scientists and non-scientists. The Author Summary should immediately follow the Abstract in your revised manuscript. This text is subject to editorial change and should be distinct from the scientific abstract. Please see our author guidelines for more information: https://journals.plos.org/plosmedicine/s/revising-your-manuscript#loc-author-summary.

The summary should include 2-3 single sentence, individual bullet points under each of the questions. The last bullet point should describe the main limitation(s) of the study's methodology.

It may be helpful to review currently published articles for examples which can be found on our website here https://journals.plos.org/plosmedicine/

INTRODUCTION

Please remove the study results and/or conclusion from the Introduction and conclude the Introduction with a clear description of the study question or hypothesis.

METHODS AND RESULTS

Please ensure that the study is reported according to the RECORD guideline, and include the completed RECORD checklist as Supporting Information. When completing the checklist, please use section and paragraph numbers, rather than page numbers. Please add the following statement, or similar, to the Methods: "This study is reported as per the REporting of studies Conducted using Observational Routinely-collected health Data (RECORD) Statement (S1 Checklist)."

Did your study have a prospective protocol or analysis plan? Please state this (either way) early in the Methods section.

For all observational studies, in the manuscript text, please indicate: (1) the specific hypotheses you intended to test, (2) the analytical methods by which you planned to test them, (3) the analyses you actually performed, and (4) when reported analyses differ from those that were planned, transparent explanations for differences that affect the reliability of the study's results. If a reported analysis was performed based on an interesting but unanticipated pattern in the data, please be clear that the analysis was data-driven.

PLOS Medicine requests that main results are quantified with 95% CIs as well as p values. When reporting p values please report as p<0.001 and where higher as the exact p value p=0.002, for example. For the purposes of transparent data reporting, if not including the aforementioned please clearly state the reasons why not.

Please include any important dependent variables that are adjusted for in the analyses.

Suggest reporting statistical information as detailed above – see under ABSTRACT

Please present numerators and denominators for percentages, at least in the Tables [not necessarily each time they're mentioned].

In your Methods section, please briefly explain what TPP is.

p.4: Please define ‘GP’.

p.4: It appears that “(Supplementary Appendix 2)” was mistakenly cited following a period. Please revise throughout your entire manuscript.

p.5: Please use a consistent spelling for the word ‘socio-demographic’ (versus ‘sociodemographic).

p.5: Please define ‘IMD’ at first use and briefly explain what IMD is.

p.5: For ”δ-change >= 1·84kg/m2/year)”, please use a greater-than-or-equal sign and revise throughout your entire manuscript (including the Supplementary material).

p.10: Please define ‘SD’, ‘aOR’.

p.10: Please, following ‘younger adults’ add an age range in brackets for clarity.

DISCUSSION

p.13: Please define ‘MNAR’. 

TABLES

Please note the use of commas to separate upper and lower bounds, as opposed to hyphens as these can be confused with reporting of negative values. Suggest reporting statistical information as detailed above – see under ABSTRACT.

Please define abbreviations used in the tables (including those in Supporting Information files).

Table 1: Please define ‘IMD’, ‘TIA’.

Table 1: It appears you have included the table title and the definitions for the asterisks (table description) twice. Please check and revise.

Table 1: Why is the oldest age group defined as ’80-89’ but the table description mentions that “Data were extracted on all adults aged >18 to ≤ 90 years […]”? This would mean the oldest age group would need to be defined as ’80-90’. Please revise throughout your entire manuscript.

FIGURES

For all Figures, please ensure that you have complied with our figures requirements http://journals.plos.org/plosmedicine/s/figures.

Please define abbreviations used in the figure legend of each figure and/or table (including those in Supporting Information files).

Please consider avoiding the use of red and green in order to make your figure more accessible to those with colour blindness.

Figure 1: Please define ‘GP’, ‘IMD’, ‘kg’, ‘m’.

Figure 2: Please add a unit for ‘age group’. Please define ‘TIA’, ‘IQR’.

Figure 2: Please indicate in the figure caption the meaning of the whiskers and the dashed vertical line. Please add an axis description. 

Figure 3: Please define ‘IMD’, ‘CI’, ‘TIA’

Figure 3: For consistency, we suggest changing the title to “Prepandemic and pandemic estimated Risk of Rapid Weight Gain (> 0.5 kg/m2/year) amongst adults living in England” or similar. Please indicate in the figure caption the meaning of the whiskers.

Figur

---

## [Decision Letter · Decision Letter 2]

10 Jan 2024

Dear Dr. Samuel,

Thank you very much for submitting your manuscript "Trends in weight gain recorded in English primary care before and during the Coronavirus-19 pandemic: an observational cohort study using the OpenSAFELY platform" (PMEDICINE-D-23-01514R2) for consideration at PLOS Medicine. 

Thank you for your detailed response to the editors' and reviewers' comments. In general, the changes made to the paper were satisfactory to the reviewers. However, as you will see, we invited a new statistical reviewer, who raised some concerns about the statistical approach and points of clarification that we feel must be addressed before we can make a final decision regarding publication. We sought a new statistical review because the first one did not provide the level of detail that we expect, and of course it is important that we ensure that all manuscript receive a thorough statistical review. Based on the new review and after discussing the paper with my colleagues and the Academic Editor, we ask you to address the comments in a further revision. We understand that this might be frustrating, but we hope you understand our reasoning. When you submit your revised paper, please include a detailed point-by-point response to the editorial comments.

The reviews are appended at the bottom of this email and any accompanying reviewer attachments can be seen via the link below:

[LINK]

In light of these reviews, I am afraid that we will not be able to accept the manuscript for publication in the journal in its current form, but we would like to consider a revised version that addresses the reviewers' and editors' comments. Obviously we cannot make any decision about publication until we have seen the revised manuscript and your response, and we plan to seek re-review by one or more of the reviewers. 

We expect to receive your revised manuscript by Jan 31 2024 11:59PM. Please email me (aschaefer@plos.org) if you have any questions or concerns.

We look forward to receiving your revised manuscript. 

Sincerely,

Alexandra Schaefer, PhD

PLOS Medicine

plosmedicine.org

Please take a look at the statistical review by reviewer #4 and respond to the comments.

Comments from the reviewers:

Reviewer #2: Authors have sufficiently addressed my comments. Thank you!

Reviewer #4: See attachment

Michael Dewey

[LINK]

---

## [Decision Letter · Decision Letter 3]

13 Mar 2024

Dear Dr. Samuel,

Thank you very much for re-submitting your manuscript "Trends in weight gain recorded in English primary care before and during the Coronavirus-19 pandemic: an observational cohort study using the OpenSAFELY platform" (PMEDICINE-D-23-01514R3) for review by PLOS Medicine.

Thank you for your detailed response to the editors' and reviewers' comments. I have discussed the paper with my colleagues and the academic editor, and it has also been seen again by the statistical reviewer. As you may notice, the statistical reviewer reiterates his concern regarding using a model which is not adjusted for other covariates. We have discussed this point extensively among the editors and, although we are sensitive to the fact that we had initially suggested that you adjust only for age and sex, we found the statistical reviewer’s comments compelling and have decided that a fully adjusted analysis should be the main analysis presented in the manuscript. We realize that this may be frustrating, but we are convinced that the fully adjusted analysis will result in a more robust presentation and analysis of the study question. 

Regarding the statistical reviewer's comments, we feel that providing the fully adjusted analyses will not require substantial additional work, given that you already present this analysis in figure S1; rather it will only require a rearrangement of the existing analyses already included in your manuscript. We apologize for the inconvenience this may cause and for the misalignment of the input from the Academic Editor, the comments from the statistical reviewers, and our Editorial input and guidance. We hope you understand our reasoning for this decision. Certainly, we are happy to give you additional time for revisions, as we understand that reorganizing your analyses may take more time.

I’m pleased to tell you that we intend to accept the paper for publication, provided you address the reviewer and editorial comments below in a further revision. When submitting your revised paper, please again include a detailed point-by-point response to the editorial comments.

[LINK]

In revising the manuscript for further consideration here, please ensure you address the specific points made by each reviewer and the editors. In your rebuttal letter you should indicate your response to the reviewers' and editors' comments and the changes you have made in the manuscript. Please submit a clean version of the paper as the main article file. A version with changes marked must also be uploaded as a marked up manuscript file. Please also check the guidelines for revised papers at http://journals.plos.org/plosmedicine/s/revising-your-manuscript for any that apply to your paper.

We ask that you submit your revision within 1 week (Mar 20 2024). As mentioned above, please contact me by email if you need additional time, and we can discuss a suitable alternative.

Please do not hesitate to contact me directly with any questions (aschaefer@plos.org). If you reply directly to this message, please be sure to 'Reply All' so your message comes directly to my inbox.

We look forward to receiving the revised manuscript.

Sincerely,

Alexandra Schaefer, PhD

Associate Editor 

PLOS Medicine

plosmedicine.org

Requests from Editors:

ABSTRACT

1) ll.66-70: Please change to: “Rapid pandemic weight gain was associated with sex, age, and Index of Multiple Deprivation (IMD). Male sex (male vs. female: adjusted Odds Ratio (aOR) 0.76 [95% Confidence Interval: 0.75, 0.76], p<0.001), older age (e.g. 50-59 years vs. 18-29 years: aOR 0. 60 [0.59, 0.60], p<0.001]) and living in less deprived areas (least deprived IMD quintile vs most deprived: aOR 0.80 [0.80, 0.81], p<0.001) reduced the odds of rapid weight gain.”

2) ll.66-70: We removed "(n=3,214,155)" because it might mislead readers into thinking that this is the number of people with rapid weight gain, when in fact it is the number of people with available data for the calculation.

3) ll.70-72: “Adults of White British ethnicity had higher odds of rapid weight gain than all other ethnicities (e.g. White British vs Indian: aOR X.XX [X.XX, X.XX], p<X.XXX)” - Since you are presenting the results from a White British perspective, we request that the relevant data be presented the other way around, rather than from a perspective of lower odds, as is currently done. Also, please clarify that these values refer to pandemic weight gain, not prepandemic.

AUTHOR SUMMARY

1) l.85: Please change to “Author Summary”.

2) l.99: Please change to “White British” and revise throughout the main manuscript.

3) Please rephrase the first bullet point under "What do these findings mean?" as it currently seems to repeat the results. Editorial suggestion: The COVID-19 pandemic appears to have had the greatest impact on women, young adults, and those living in the most deprived areas in terms of unhealthy patterns of weight gain.

4) The last bullet point should describe the main limitation(s) of the study's methodology only. Please revise.

METHODS AND RESULTS

1) We feel that there are currently too many subheadings that divide the text into unnecessarily many sections. Please revise. For example, “Population level trends in BMI recording activity and median BMI”, “Individual level BMI trajectories” and “Defining a population of extreme accelerators” should form one section under “BMI trajectories” or similar.

2) Similarly, please remove the subheadings under “Individual level BMI trajectory analyses” as we feel the heading is sufficient to guide the reader through the results.

3) ll.171-174: Please revert the text describing the change in analysis to focus on the minimally adjusted model, as a result of the peer review process. Modification will also be required at ll.287-288. 

4) ll.321-322: Is it worth noting that differences between prepandemic and pandemic, e.g. in the 70-79 age group, were generally quite modest.

5) l.346: Please specify in the first sentence that the results presented are for the pandemic period. 

6) l.380: Please define “Adults” and remove the duplicate “with”.

7) ll.403.404: Please change to “But there were some exceptions, e.g. among Black individuals, deprivation was not associated with weight gain (IMD5 vs IMD1: aOR 1.04[0.92,1.18], p = 0.496);…”

8) l.410: Please add “years” to “aged 60-79”.

DISCUSSION

1) l.417: Please add “to our knowledge” or similar, i.e. “We report, to our knowledge, novel findings...”.

2) ll.478-480: “Alternatively this may be a true effect arising from the awareness that COVID-19 infection confers greater risk to people with obesity and LTCs such as T2D,…” – please provide reference.

3) We suggest that the discussion should mention the problems (e.g. health problems) associated with weight gain/rapid weight gain and discuss initiatives that have been taken to address this issue (e.g., the introduction of the Soft Drink Industry Levy) in order to delve deeper into the implications and next steps for research, clinical practice, and/or public policy. 

REFERENCES

1) Where website addresses are cited, please add the word “Accessed” when specifying the date of access (e.g. [accessed: 12th February 2024]).

FIGURES

1) Figure 1: In the first box, please change to “aged 18 to ≤90 years”.

2) S1 Figure: Please be sure to define the difference between "aOR" and "aOR**" in the figure description. Assuming that the aOR** values represent the fully adjusted prepandemic values, we wonder why the minimally adjusted prepandemic aOR values are not shown, but the pandemic aOR values are? Also, the aOR** values for the prepandemic period appear to be identical to the aOR values listed under "Pandemic" - is this correct? Please revise.

3) Figure S2: The figure descriptions for Figure S2 and Figure 4 appear to be identical. Does Figure S2 show the fully adjusted values? Please revise and be sure to display two decimals in Figure S2 ("extreme acceleration") for consistency.

SOCIAL MEDIA

To help us extend the reach of your research, please provide any X (formerly known as Twitter) handle(s) that would be appropriate to tag, including your own, your co-authors’, your institution, funder, or lab. Please enter in the submission form any handles you wish to be included when we post about this paper.

Comments from Reviewers:

Reviewer #4: My main point was the use of a model which did not adjust for other covariates. I seem to be in a small minority here but I am not convinced by the rebuttal. Let us look at examples to make concrete my concern.

The well known Berkeley admissions data https://en.wikipedia.org/wiki/Simpson%27s_paradox#UC_Berkeley_gender_bias showed that women were less likely to be admitted. This was not evidence of bias since a more detailed analysis taking area of study into account, if anything, showed the reverse.

If the authors prefer examples using continuous variables we can note that men have larger brains than women (1345g v 1222 according to Wikipedia) a fact which men saw as justifying the continued subjugation of women. If we adjust for body weight (UK 85.4Kg v 72.1Kg, same source) we find the reverse.

One important thing here is that I do not think people saw these as anything other than descriptions so I find the authors insistence that they are only doing a descriptive analysis rather puzzling. Descriptions can mislead.

Michael Dewey

[LINK]

General Editorial Requests

---

## [Editor Report · Decision Letter 4]

5 Apr 2024

Dear Dr Samuel, 

On behalf of my colleagues and the Academic Editor, David Flood, I am pleased to inform you that we have agreed to publish your manuscript "Trends in weight gain recorded in English primary care before and during the Coronavirus-19 pandemic: an observational cohort study using the OpenSAFELY platform" (PMEDICINE-D-23-01514R4) in PLOS Medicine.

I appreciate your thorough responses to the reviewers' and editors' comments throughout the editorial process. We look forward to publishing your manuscript, and editorially there are only three remaining minor stylistic/presentation points that should be addressed prior to publication. We will carefully check whether the changes have been made. If you have any questions or concerns regarding these final requests, please feel free to contact me at aschaefer@plos.org.

Please see below the minor points that we request you respond to:

1) l.74: Please add “to our knowledge” or similar (i.e. “We present, to our knowledge, new evidence that people with mental health...”).

2) When reporting results, please be sure to add "95% CI" before each set of parentheses (where applicable and after introducing the abbreviation). For example, lines 309-310: “…in 18-29-year-olds (aOR 0.60, 95%CI [0.60, 0.61], p<0.001).” Similarly, please add "IQR" or other statistical details before each set of corresponding parentheses (where applicable and after introducing the abbreviation). Please revise throughout the manuscript including the abstract.

3) l.369: Please change "P<0.001" to "p<0.001".

PRESS

Sincerely, 

Alexandra Schaefer, PhD 

Associate Editor 

PLOS Medicine